# PARTNER-BENCH: EVALUATING VISUALLY-GROUNDED IQ AND INTERACTIVE EQ IN AUDIO-VISUAL DIALOGUE

## ABSTRACT

Developing Multi-Modal Large Language Models (MLLMs) from "tool-oriented" auxiliary AI to "partner-oriented" interactive AI requires a capacity for acting as an active participant in Audio-Visual Dialogue (AVD), which blends high **Visually-Grounded IQ** (the capacity to reason over joint audio-visual information) and high **Interactive EQ** (the ability to respond with empathy and expressiveness). However, progress faces a key obstacle: the absence of a unified standard for defining and evaluating a "good" AVD model. The current evaluation landscape is thus fragmented: audio dialogue benchmarks can assess Interactive EQ but remain visually blind, while audio-visual understanding benchmarks evaluate Visually-Grounded IQ but lack interactivity. This methodological gap leaves the critical synthesis of IQ and EQ in conversational contexts entirely unmeasured. To address this gap, we introduce **Partner-Bench**, the first benchmark designed to evaluate this synthesis. To construct it, we present a novel Data Engine, **Partner-DE**, which automatically mines, filters, and annotates high-quality conversational data from web videos. Partner-Bench comprises 376 samples (3.49 hours total), and features a fine-grained, 7-dimensional evaluation framework that decomposes IQ into three dimensions: Recognition, Comprehension, and Reasoning, and decouples EQ into two quality categories: Linguistic (including Persona, Cohesion) and Prosodic (including Naturalness, Affect). Our initial experiments on Partner-Bench yield three critical findings: (1) All current models perform significantly below the human baseline, indicating substantial room for improvement; (2) we observe a significant performance gap between paradigms: the current SOTA cascaded models significantly outperform existing end-to-end models (e.g., Cascaded Mimo-Audio at 68.38 vs. Qwen2.5-Omni at 51.53); (3) There is a "context cliff" where model performance initially improves with longer context but then sharply declines, indicating a failure to process extended interactions. By providing a rigorous standard and a diagnostic tool to pinpoint such weaknesses, Partner-Bench aims to steer the improvement of AVD models and ultimately accelerate the development of the next generation of truly perceptive and engaging AI companions.

## 1 INTRODUCTION

The trajectory of Multi-Modal Large Language Models (MLLMs) is marked by a fundamental paradigm shift (OpenAI et al., 2024; Fu et al., 2025): from passive, "tool-oriented" assistants (e.g., search engines, image retrieval, problem-solving, coding) executing discrete commands to proactive, "partner-oriented" companions (e.g., AI partners, embodied agents, AI butlers, customer service bots) capable of engaging in the rich tapestry of human interaction. A core bridge for this transition is proficiency in Audio-Visual Dialogue (AVD), a task that necessitates a synthesis of two distinct capabilities. The first is Visually-Grounded Intelligence Quotient (IQ), which involves the ability to accurately perceive, comprehend, and reason about the visual world in conjunction with auditory information. The second is Interactive Emotional Quotient (EQ), which encompasses the ability to communicate naturally, empathetically, and coherently as an active dialogue participant. Equipped with these two key capabilities, AI can truly understand and respond to human emotions and intentions in the physical world, thereby enabling deeper communication and connection.

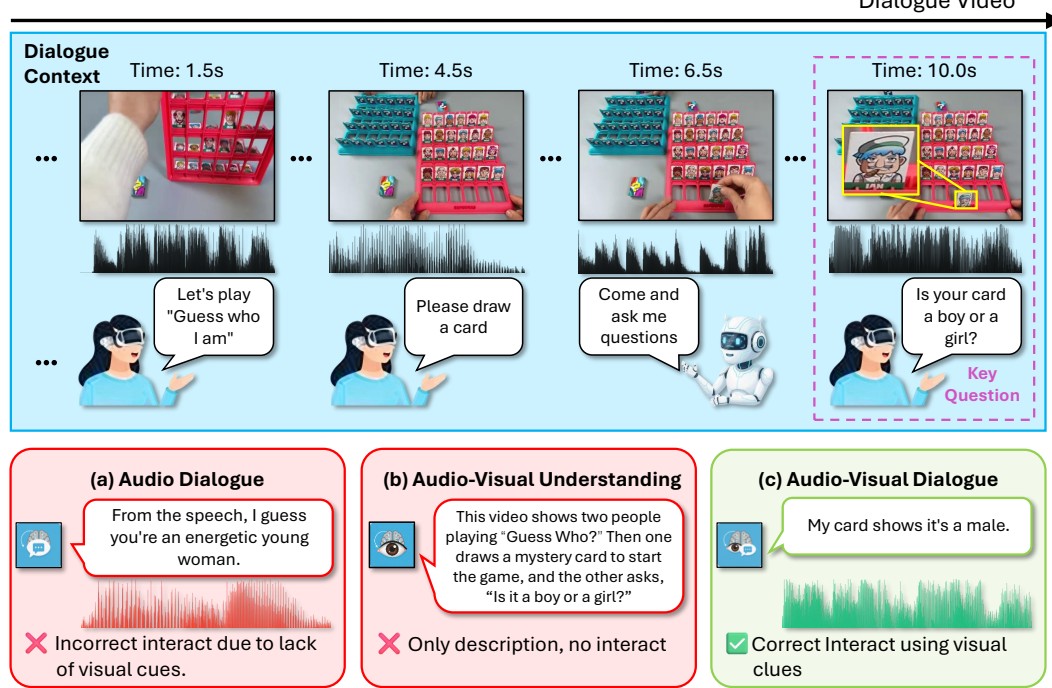

Figure 1: **Comparison of Audio-Visual Dialogue with preceding paradigms.** (a) **Audio Dialogue**: High EQ, but being visually blind leads to ungrounded or factually incorrect responses. (b) **Audio-Visual Understanding**: High IQ, but the lack of interactivity makes them passive observers, not participants. (c) **Audio-Visual Dialogue** (Our Goal): Synthesizes IQ and EQ, using visual cues to ground a contextually appropriate, interactive response for meaningful communication.

Despite this exciting vision and the current progress made in audio-visual dialogue models (Xu et al., 2025; Zhang et al., 2025), progress towards truly capable AVD systems is still hampered by a fundamental obstacle: the absence of a unified and widely accepted standard for defining and evaluating what constitutes a "good" AVD model. This lack of a standardized evaluation protocol not only creates ambiguity in research objectives but also acts as a critical bottleneck, slowing the iterative cycle of development and improvement. Consequently, advancing from current models to the next generation of perceptive AI partners requires, first and foremost, the establishment of a rigorous and comprehensive evaluative benchmark.

The current evaluation landscape is bifurcated, forcing the advanced AVD models (e.g., Qwen2.5-Omni (Xu et al., 2025) and Stream-Omni (Zhang et al., 2025)) to be assessed on benchmarks that address the components of AVD in isolation but not their synthesis, which fails to capture the full scope of AVD capabilities. On one hand, if assessed on **Audio Dialogue benchmarks** (e.g., SD-Eval (Ao et al., 2024), URO-Bench (Yan et al., 2025)), a powerful AVD model is reduced to a visually-impaired agent. As depicted in Figure 1(a), while it may exhibit high **Interactive EQ** by maintaining conversational coherence, its responses are fundamentally ungrounded. Being **visually blind**, it is prone to making factually incorrect or nonsensical statements whenever the dialogue hinges on visual context. On the other hand, if assessed on **Audio-Visual Understanding benchmarks** (e.g., OmniBench (Li et al., 2024), AV-Odyssey (Gong et al., 2024b)), the model is relegated to the role of a passive observer. As shown in Figure 1(b), it can demonstrate high **Visually-Grounded IQ** by accurately describing scenes and events. However, these benchmarks almost entirely fail to evaluate a model's ability to assume a persona and engage in a highly interactive dialogue, which is central to its EQ. Consequently, the critical nexus where visual reasoning and conversational acumen must converge—the scenario depicted in Figure 1(c) and the very essence of a perceptive AI partner—remains an entirely unmeasured and unbenchmarked territory.

To address this critical evaluation gap, we introduce **Partner-Bench**, the first benchmark designed to evaluate the holistic synthesis of Visually-Grounded IQ and Interactive EQ in AVD. To construct it, we present a novel and scalable Data Engine, **Partner-DE**, which automatically mines, filters,

Table 1: **Partner-Bench vs. Existing Benchmarks: IQ/EQ Evaluation Framework Comparison**. IQ is decomposed into three dimensions: Recognition (RECOG.), Comprehension (COMP.), Reasoning (REAS.). EQ is decoupled into two categories: Linguistic (Persona, Cohesion) and Prosodic (Naturalness, Affect). Modality notation: A (Audio), V (Video), I (Image), T (Text). Audio Dialogue benchmarks are visually blind (focus solely on EQ), while most Audio-Visual Understanding benchmarks are conversationally deaf (neglect EQ entirely). Partner-Bench is the first to enable unified evaluation across all IQ/EQ dimensions for audio-visual dialogue agents (A+V+T → A).

| Benchmark | Modality | Visually-Grounded IQ | | | Interactive EQ | |
|---|---|---|---|---|---|---|
| | | RECOG. | COMP. | REAS. | Ling. | Pros. |
| Audio Dialogue | | | | | | |
| SD-Eval | A+T → T | ✗ | ✗ | ✗ | ✔ | ✗ |
| URO-Bench | A+T → A | ✗ | ✗ | ✗ | ✔ | ✔ |
| MTalk-Bench | A+T → A | ✗ | ✗ | ✗ | ✔ | ✔ |
| Audio-Visual Understanding | | | | | | |
| AVQA | A+V+T → T | ✔ | ✔ | ✗ | ✗ | ✗ |
| Music-AVQA | A+V+T → T | ✔ | ✗ | ✗ | ✗ | ✗ |
| RefAVS-Bench | A+V+T → T | ✔ | ✗ | ✗ | ✗ | ✗ |
| AVHBench | A+V+T → T | ✔ | ✔ | ✗ | ✗ | ✗ |
| CMM | A+V+T → T | ✔ | ✗ | ✗ | ✗ | ✗ |
| OmniBench | A+I+T → T | ✔ | ✔ | ✔ | ✗ | ✗ |
| AV-Odyssey | A+I/V+T → T | ✔ | ✔ | ✔ | ✗ | ✗ |
| Audio-Visual Dialogue | | | | | | |
| PartnerBench | A+V+T → A | ✔ | ✔ | ✔ | ✔ | ✔ |

and annotates high-quality audio-visual conversational scenarios from web videos. Partner-Bench comprises 376 samples (3.49 hours total). It decomposes Visually-Grounded IQ into three dimensions: Recognition, Comprehension, and Reasoning, and decouples Interactive EQ into two major categories: Linguistic (encompassing Persona and Cohesion) and Prosodic (including Naturalness and Affect). This comprehensive seven-dimensional framework enables a more granular and holistic assessment than prior works. As shown in Table 1, this stands in stark contrast to existing audio dialogue and audio-visual understanding benchmarks, which typically cover only a narrow subset of these essential capabilities.

Our initial experiments on Partner-Bench, evaluating a range of state-of-the-art models from cascaded to end-to-end audio-visual schemes, yield several critical findings. We observe that: (1) Current models perform significantly below the human baseline across all dimensions, indicating substantial room for improvement; (2) There exists a pronounced performance gap between different modeling paradigms: the current cascaded systems significantly outperform their end-to-end counterparts (e.g., a top score of 68.38 vs. 51.53). (3) More strikingly, we identify a "context cliff": a phenomenon where model performance, after initially improving with longer conversational context, sharply declines past a certain threshold, indicating a systemic failure in processing extended interactions. By providing a rigorous standard to surface such weaknesses and a diagnostic tool to analyze them, Partner-Bench aims to steer targeted improvements and ultimately accelerate the development of the next generation of truly perceptive and engaging AI companions.

## 2 RELATED WORKS

Our work is positioned at the intersection of speech dialogue and audio-visual understanding. Table 1 details the comparison of Partner-Bench with existing works via our IQ/EQ framework.

**Speech Dialogue.** Benchmarks like SD-Eval (Ao et al., 2024), URO-Bench (Yan et al., 2025), and MTalk-Bench (Du et al., 2025) have been instrumental in evaluating **Interactive EQ**, focusing on conversational coherence and naturalness. However, as shown in Table 1, they are fundamentally **visually blind** and entirely neglect the evaluation of **Visually-Grounded IQ**.

**Audio-Visual Understanding.** Conversely, benchmarks such as AVQA (Yang et al., 2022), Music-AVQA (Guangyao et al., 2022), Ref-AVS (Wang et al., 2024), AVHBench (Sung-Bin et al., 2024),

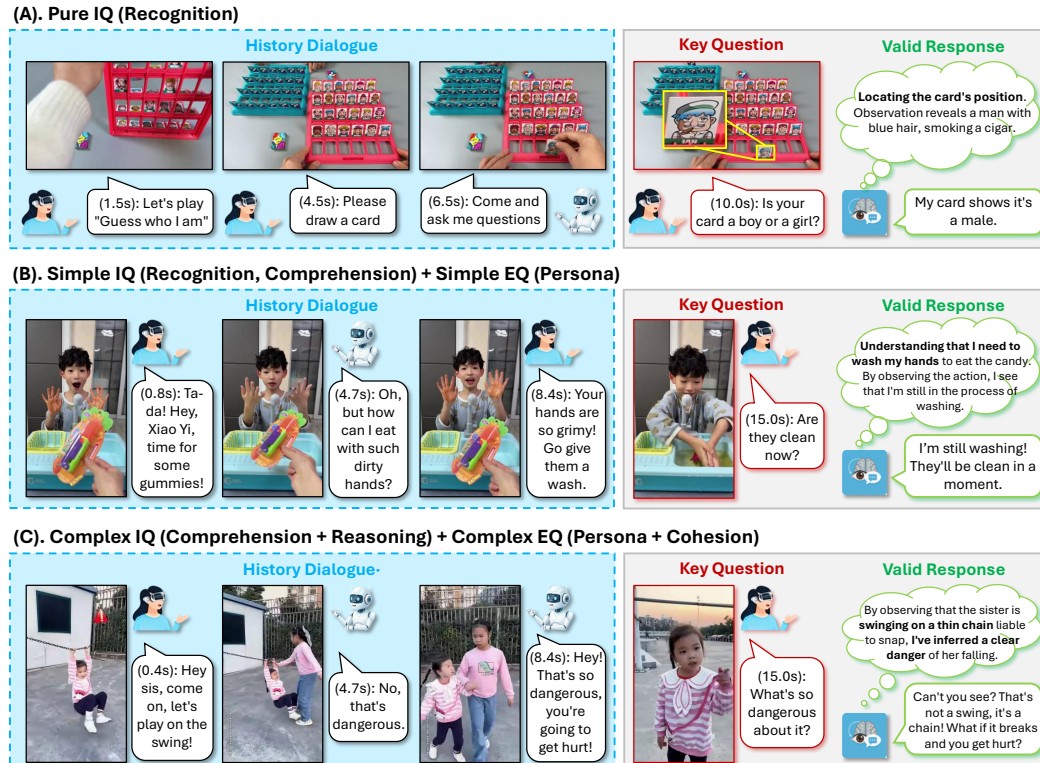

Figure 2: **Exemplars from Partner-Bench: Tasks with Progressive Complexity (A) Pure IQ:** A basic *Recognition* task that requires identifying visual attributes. **(B) Simple IQ + EQ:** A task that demands both event *Comprehension* (e.g., washing hands before eating) and maintenance of a consistent *Persona*. **(C) Complex IQ + EQ:** An advanced scenario requiring visually-grounded *Reasoning* to infer non-obvious dangers (e.g., a thin swinging chain), and expression of the inference with conversational *Cohesion*. Thought bubbles visualize the internal inference process.

CMM (Leng et al., 2024), OmniBench (Li et al., 2024), and AV-Odyssey (Gong et al., 2024a) provide strong evaluation for **Visually-Grounded IQ**, particularly in Recognition and Comprehension. Nonetheless, these benchmarks are typically single-turn and non-interactive, thus failing to assess a model's **Interactive EQ**, as detailed in Table 1.

**Audio-Visual Dialogue.** The nascent field of AVD aims to synthesize the capabilities from both domains. While powerful models like Qwen2.5-Omni (Xu et al., 2025) have emerged, their evaluation is hindered by the lack of a suitable benchmark. Existing works either repurpose AV understanding benchmarks, thereby ignoring EQ, or focus on narrow sub-tasks. In contrast, **Partner-Bench is the first to provide a unified platform that comprehensively evaluates all dimensions of our proposed IQ/EQ framework**, establishing a much-needed, holistic standard for the field.

## 3 THE FRAMEWORK OF PARTNER-BENCH

In this section, we detail the comprehensive evaluation framework of Partner-Bench, designed to rigorously assess the dual capabilities of Visually-Grounded IQ and Interactive EQ in AVD models. We first define the fine-grained dimensions that constitute these capabilities in Section 3.1, and then introduce the protocol for evaluating models against these dimensions in Section 3.2.

### 3.1 EVALUATION DIMENSIONS

In real-world audio-visual dialogue scenarios, AI must simultaneously demonstrate high-level Visually-Grounded IQ and Interactive EQ. These two capabilities are not isolated; rather, they are complementary and inseparable, collectively forming the core competencies required for AI to func-

tion as a conversational partner, as illustrated in Figure 2. To move beyond monolithic scores and enable granular diagnostics, we decompose the complex skillset of AVD into a multi-dimensional challenge profile:

**Visually-Grounded IQ**. We model a AVD model's intelligence as a three-tiered cognitive chain, progressing from perception to reasoning: (1) **Recognition (RECOG.):** This is the foundational ability to "see" clearly by identifying static visual facts. For example, in Figure 2 (A), the model must accurately recognize the visual attributes of the selected character card, such as gender, to provide a valid response. (2) **Comprehension (COMP.):** This is the ability to "understand" dynamic context by linking discrete facts into coherent events. In Figure 2 (B), for instance, a valid response requires the model to comprehend the ongoing process of the character washing their hands. (2) **Reasoning (REAS.):** This represents the highest cognitive "thinking" ability: inferring causality or solutions by synthesizing visual percepts with world knowledge. As shown in Figure 2 (C), the model must go beyond simple observation to reason about the non-obvious danger of a faulty swing, a conclusion requiring real-world inference.

**Interactive EQ**. We innovatively decouple a model's expressiveness into its linguistic content and prosodic delivery. (1) **Linguistic EQ (EQ-L.):** This dimension assesses the quality of the textual "script." It encompasses attributes such as: ❶ *Persona Consistency:* The ability to adopt a specific role rather than merely describing facts, as required in the scenarios of Figure 2 (B) and (C). ❷ *Cohesion & Awareness:* The capacity to generate responses that are not only logically coherent but also contextually aware of the conversational atmosphere. For instance, in Figure 2 (C), the response must be tailored to the established "dangerous" context, rather than being a neutral, detached statement. (2) **Prosodic EQ (EQ-P.):** This dimension evaluates the quality of the audio "performance." It includes: ❶ *Naturalness & Fluency:* The intrinsic quality of the synthesized speech, such as its resemblance to human speech patterns, which is a universally applicable attribute for all samples. ❷ *Affective Appropriateness:* The alignment of vocal emotion and intonation with the linguistic content. This attribute is the acoustic counterpart to linguistic cohesion; for a response to be truly effective, the "performance" must match the "script."

## 3.2 EVALUATION PROTOCOL

Based on our defined dimensions, we establish a rigorous and scalable protocol for evaluation that simulates real-world, context-dependent interaction.

**Task Formulation and Input.** We formally define a dialogue in a video as a sequence of turns, $D = [d_1, d_2, ..., d_n]$, where each turn $d_i$ is a tuple $\{t_{si}, t_{ei}, s_i, c_i, a_i\}$ representing the start time, end time, speaker, textual content, and source audio of the utterance. To simplify the task, we process the data into a dyadic dialogue, i.e., speaker$\in \{A, B\}$. To evaluate a model's ability to generate a response for a target turn $d_i$, we adopt a contextual window approach. The model under test (MUT) is provided with two inputs: (1) **A system prompt** instructing the model to act as an in-context dialogue participant (if the questioner is Speaker A, the model shall act as Speaker B, and vice versa.), grounding its response in both visual information and conversational semantics. (2) **An audio-visual clip** spanning the time interval $[\max(0, t_{si} - w), t_{ei}]$, where $w$ is the context window duration. $[\max(0, t_{si} - w), t_{si}]$ is the history context, and $[t_{si}, t_{ei}]$ is the target turn for which the model must generate a response. The MUT assumes the role of either Speaker A or B in the dialogue context to generate a spoken response (audio signal $a_i'$, with transcribed content $c_i'$). This response must appropriately and coherently continue the dialogue from $d_i$, and the evaluation then assesses the quality of both $c_i'$ and $a_i'$.

**Method: Model-as-Judge with a Dimensional Scorecard.** Given the generative nature of the task, we employ a model-as-judge approach for scalable evaluation. Specifically, we use Gemini-2.5-Pro as the Referee model. For IQ and EQ-L., we prompt the Referee to assign scores using three inputs: the complete context (including the audio-visual clip and dialogue history), the role that the model needs to embody, the reference ground truth response ($c_i^{ref}$), and the generated response from the MUT ($c_i'$). For EQ-P., since its evaluation focuses on audio quality, no reference audio is provided; instead, we only input the complete context and the spoken response $a_i'$ generated by the MUT for auditory assessment. The MUT's performance on each relevant IQ and EQ dimension is initially rated on a 5-point Likert scale, which we then multiply by 20 to convert to a 100-point scale.

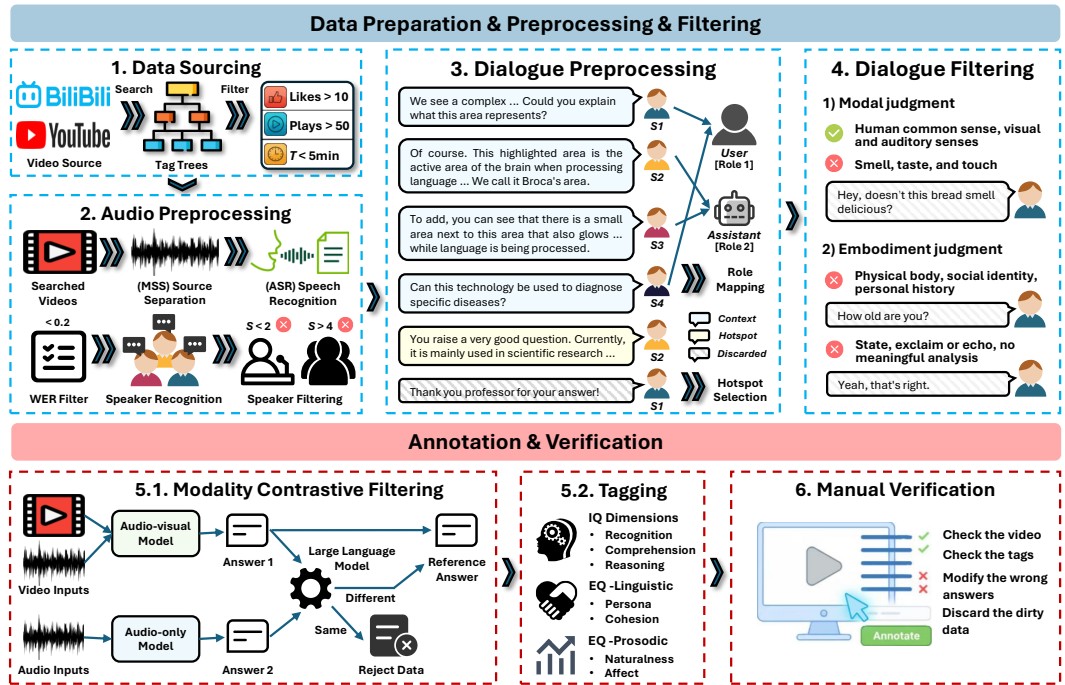

Figure 3: **The Partner-DE pipeline for supporting Partner-Bench.** The process consists of two main phases. **Phase 1 (Steps 1-4)** efficiently acquires dialogue candidates from massive video volumes and converts them into structured two-party dialogue texts with speaker information. **Phase 2 (Steps 5-6)** transforms these dialogue texts into final evaluation units. specifically, AVD samples with multi-dimensional challenge labels and high-quality reference answers.

## 4 THE DATA ENGINE OF PARTNER-BENCH

Manually curating a high-quality benchmark for audio-visual dialogue that robustly evaluates both IQ and EQ is prohibitively labor-intensive and costly. Therefore, we present our Data Engine, a comprehensive pipeline designed to automatically mine, process, and annotate high-quality, multi-turn conversational scenarios from massive volumes of web videos, thereby enabling the scalable construction of Partner-Bench, as shown in Figure 3.

### 4.1 PHASE 1: DATA PREPARATION, PREPROCESSING, AND FILTERING

The first phase of our engine is dedicated to transforming a vast corpus of raw web videos into a refined set of structured, high-potential dialogue candidates. This is achieved through the following four steps, as illustrated in the upper half of Figure 3.

**Step 1: Data Sourcing.** We begin by sourcing candidate videos from major platforms like YouTube and BiliBili. To maximize the probability of finding relevant conversational content, we employ a domain-targeted search strategy guided by a hierarchical topic tree ("Domain -> Topic -> Keywords", Domain and Topic can refer to Figure. A1 (a)). The retrieved videos are then subjected to an initial quality filter based on engagement metrics (e.g., "Likes > 10", "Plays > 50") and duration (e.g., "T < 5min") to discard low-quality or overly long content.

**Step 2: Audio Preprocessing.** Next, the audio track of each video is processed through a multi-stage pipeline to extract clean, transcribed speech. We first apply Music Source Separation (MSS) to mitigate background noise. The resulting speech signal is then processed by a Voice Activity Detection (VAD) model and a state-of-the-art Automatic Speech Recognition (ASR) system to generate time-stamped utterances. To ensure transcription quality, we discard any video with a Word Error Rate (WER) exceeding 0.2. Finally, a speaker recognition model assigns a speaker ID to each utterance. Dialogues with fewer than two or more than four speakers are filtered out to maintain a focused, analyzable conversational structure. The final transcripts follows the same format as that in 3.2; however, $s_i \in \{2, 3, 4\}$, meaning the number of speakers is variable.

**Step 3: Dialogue Preprocessing.** The structured transcripts then undergo preprocessing to prepare them for evaluation. First, we normalize the speaker IDs ($s_i \in \{2, 3, 4\}$) by consolidating them into a two-party ("speakerA", "speakerB") format dialogue ($s_i \in \{A, B\}$), to standardize the interactive scenario. Subsequently, we identify and label high-quality, information-seeking dialogue turns as **"Hotspots."**. These hotspots represent potential trigger prompts for our benchmark. Turns with low informational value, such as simple acknowledgments, declarative statements, or fragmented utterances, are discarded as they are unsuitable for forming a meaningful evaluation unit.

**Step 4: Dialogue Filtering.** The identified hotspots are subjected to a final, crucial round of quality filtering to ensure they are fair and evaluable for a disembodied AI. We prompt a LLM to apply two judgment filters to the hotspots. The first is a **Modal Judgment** filter, which discards questions that require senses beyond vision and audition, such as taste or smell. The second is an **Embodiment Judgment** filter, which removes questions that presuppose the model has a physical body, personal attributes (e.g., height, age), or the ability to perform physical actions. This step ensures that all candidate hotspot questions are solvable through audio-visual perception and reasoning alone.

### 4.2 Phase 2: Data Annotation and Verification

Following the initial preparation and filtering, the refined set of dialogue candidates undergoes a final, intensive phase of annotation and verification to produce the benchmark units. This phase ensures that each sample is not only visually dependent but also enriched with a detailed challenge profile and a high-quality reference answer, as illustrated in the lower half of Figure 3.

**Step 5.1: Modality Contrastive Filtering.** A key requirement for our benchmark is that hotspot questions must **require** visual information for correct responses. To enforce this, we introduce a novel **Modality Contrastive Filtering** step. As shown in Figure 3 (5.1), for a given question, we query two models: a full audio-visual model (Answer 1) and an audio-only model (Answer 2). These two answers are then compared by a powerful MLLM (Gemini-2.5-Pro). If judged the same or highly similar, we conclude visual information is redundant; such samples are deemed "audio-only" questions and rejected, ensuring all retained data has strong visual grounding. Retained "Answer 1"s are preserved as a preliminary reference for the next step.

**Step 5.2: Dimensional Challenge Tagging.** Once a sample's visual dependency is confirmed, we proceed to annotate it with a detailed challenge profile. We employan a MLLM-based annotator (suitable prompt + Gemini-2.5-pro), to analyze the question in its full audio-visual context. As shown in Figure 3 (5.2), this step automatically tags each sample with all the specific **IQ Dimensions** (Recognition, Comprehension, Reasoning) and **EQ Dimensions** (Linguistic: Persona, Cohesion; Prosodic: Naturalness, Affect) that are significantly challenged by the task.

**Step 6: Human Verification.** Our team of human experts meticulously verifies the following: whether the video quality meets standards, whether the dialogue and questions are appropriate, whether the assigned challenge dimensions are accurate, and they refine any potentially erroneous reference answers. Only samples that pass this final comprehensive review are included in the Partner-Bench dataset, thus ensuring the dataset serves as a gold-standard evaluation tool.

## 5 Experiments

### 5.1 Experimental Setup

**Dataset Statistics.** Our experiments are conducted on our constructed Partner-Bench, which contains 376 high-quality audio-visual dialogue units from 294 distinct videos. For the detailed breakdown of sample counts by video duration and the distribution of samples across different evaluation dimensions, please refer to Figure A1 (c) and (d).

**Metrics.** The primary evaluation and scoring methods are defined in Section 3.2. Notably, the calculation of the Overall score follows $s_{\text{overall}} = s_{\text{IQ}} \times 0.5 + s_{\text{EQ-L}} \times 0.25 + s_{\text{EQ-P}} \times 0.25$, where $s_{\text{IQ}}$, $s_{\text{EQ-L}}$, and $s_{\text{EQ-P}}$ are the average scores across all relevant dimensions within each capability pillar.

**Evaluated Models.** We evaluate a comprehensive set of models, categorized into three paradigms, and compare them against a **Human Baseline** (representing the performance upper bound). Specifically: (1) **Cascaded (Cap2Dial):** Video Captioning model (e.g., Gemini-2.5-pro (Comanici

Table 2: **Main Results on Partner-Bench.** Video Captions are generated by Gemini-2.5-Pro ("*" denotes captions are generated by Qwen2.5-VL (7B)). TTS model is MeloTTS.

| Model | Visually-Grounded IQ | | | Interactive EQ-L. | | Interactive EQ-P. | | Overall |
|---|---|---|---|---|---|---|---|---|
| | RECOG. | COMP. | REAS. | Persona | Cohesion | Natural | Affect | |
| Human | 85.72 | 87.85 | 71.49 | 74.98 | 74.72 | 85.34 | 65.97 | 78.71 |
| **Cascade (Cap2Dial):** Video Caption Model w/ Speech-to-Speech Model | | | | | | | | |
| Kimi-Audio (7B) | 58.68 | 62.74 | 45.54 | 58.81 | 57.11 | 79.37 | 67.17 | 60.63 |
| Step-Audio (140B) | 64.40 | 67.95 | 51.88 | 59.23 | 59.50 | 80.74 | 66.54 | 63.96 |
| Mimo-Audio (7B) | 61.45 | 66.85 | 56.53 | 62.98 | 66.54 | 84.92 | 86.16 | 68.38 |
| *Mimo-Audio (7B) | 40.82 | 49.95 | 41.68 | 50.64 | 49.18 | 82.91 | 72.96 | 54.04 |
| **Cascade (AVQA2TTS):** (Audio-Visual Model w/ TTS Model) | | | | | | | | |
| VideoLLaMA2 (7B) | 27.11 | 24.84 | 20.10 | 20.00 | 20.00 | 52.22 | 20.13 | 26.05 |
| VITA-1.5 (8B) | 44.53 | 43.93 | 30.20 | 28.51 | 27.17 | 66.14 | 36.73 | 39.59 |
| Gemini-2.5-Flash (-) | 59.12 | 63.65 | 42.87 | 62.13 | 53.33 | 81.43 | 58.36 | 59.51 |
| Gemini-2.5-Pro (-) | **66.98** | **75.43** | 58.71 | 75.49 | 71.45 | 78.99 | 66.42 | 70.06 |
| **End-to-End** | | | | | | | | |
| Stream-Omni (8B) | 19.94 | 19.82 | 19.80 | 19.66 | 19.75 | 54.02 | 20.00 | 24.10 |
| Qwen2.5-Omni (7B) | 49.62 | 48.86 | 38.61 | 44.94 | 46.42 | 85.29 | 52.83 | 51.53 |

et al., 2025), Qwen2.5-VL (Bai et al., 2025)) with an Audio Dialogue LLM for dialogue generation (e.g., Kimi-Audio (KimiTeam et al., 2025), Step-Audio (Huang et al., 2025), Mimo-Audio (Xiaomi, 2025)). (2) **Cascaded (AVQA2TTS):** Audio-Visual Understanding model (e.g., VideoLLaMA2 (Cheng et al., 2024), VITA-1.5) to generate text, followed by a TTS model (e.g., MeloTTS (Zhao et al., 2023)). (3) **End-to-End:** single-architecture models that jointly process all modalities, including **Qwen2.5-Omni** (Xu et al., 2025) and **Stream-Omni** (Zhang et al., 2025).

## 5.2 MAIN RESULTS

We present the main evaluation results on Partner-Bench in Table 2. It details the performance of all evaluated models across all seven fine-grained IQ and EQ dimensions, as well as the aggregated Overall score. Human performance is also included to serve as a practical upper bound. Our experiments yield three critical and revealing findings regarding the current state of AVD models:

**Finding 1: A Significant Gap to Human Performance.** As shown in Table 2, the top-performing system, a cascaded Gemini-2.5-Pro, still underperforms the Human Baseline (70.06 vs. 78.71), particularly in terms of IQ (generally lagging by 10–20 points). However, in terms of EQ, the current best cascaded model performs well and can approach the Human Baseline.

**Finding 2: The Dominance of Cascaded Paradigms.** The top-performing cascaded systems, particularly those in the "Cap2Dial" category (e.g., Mimo-Audio at 68.38) and the top-tier "AVQA2TTS" models (e.g., Gemini-2.5-Pro at 70.34), significantly outperform the end-to-end models (e.g., Qwen2.5-Omni at 51.53) by a large margin of over 17-18 points, which is more intuitively visible in Figure A2. It is also noteworthy that even when using the Qwen2.5-VL-7B model as the captioning front-end for Mimo-Audio, the resulting cascaded system still surpasses the end-to-end Qwen2.5-Omni (54.04 vs 51.53).

**Finding 3: A Universal Weakness in IQ dims.** In the IQ domain, nearly all models—including humans, find **Reasoning (REAS.)** the most challenging dimension, indicating that high-level inference remains a major bottleneck. Furthermore, compared to humans, models exhibit substantial gaps across all IQ dimensions (e.g., Best model, Gemini-2.5-Pro still lags by 18.74, 12.43, and 12.78 points in RECOG., COMP., and REAS., respectively). This indirectly reflects the current limitations of models in visual understanding and multimodal reasoning within dialogue scenarios.

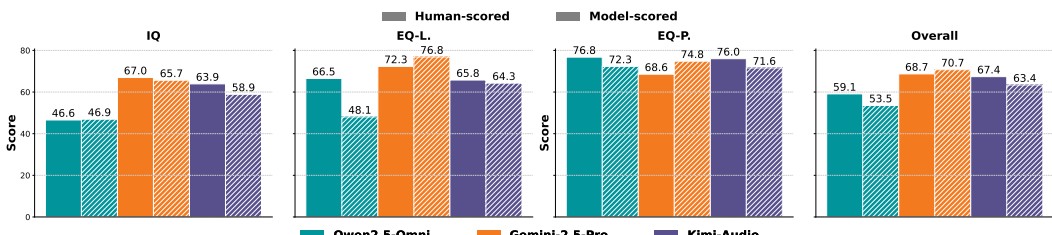

Figure 4: **Correlation between model-judge scores and human-judge scores.**

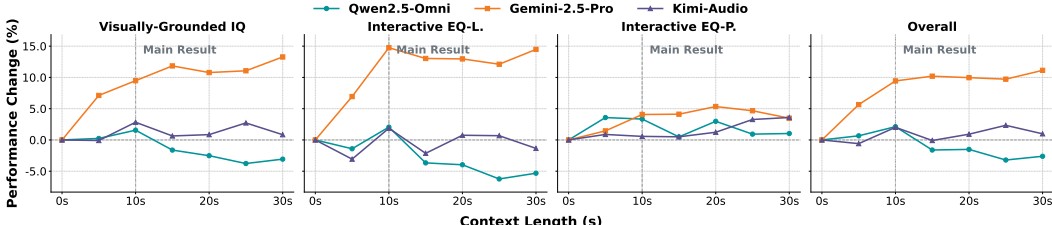

Figure 5: **The "Context Cliff" phenomenon of end-to-end model.** As the context length $w$ increases, model performance initially improves but then sharply declines after a certain point, indicating a systemic failure in handling long-range interactions.

### 5.3 FURTHER ANALYSIS: PROTOCOL RELIABILITY AND CONTEXT EFFECTS

**Correlation between our Evaluation Framework and Human Judgement.** We randomly sampled 100 dialogue units and collected responses from three representative models. These responses were then scored in parallel by both human annotators ("Human-scored") and our Gemini-2.5-Pro Referee ("Model-scored") using the identical dimensional scorecard. The results, presented in Figure 4, demonstrate a strong and consistent alignment between the two scoring methods. For instance, in the "Overall" assessment, the human-scored rating for Gemini-2.5-Pro is 68.7, which is highly comparable to the model-scored rating of 70.7; similarly, for Kimi-Audio, the scores are 67.4 (human) and 63.4 (model). These results robustly validate our model-as-judge protocol, confirming that our automated Referee serves as a reliable and consistent proxy for human evaluation, enabling the large-scale and cost-effective assessment of AVD model capabilities.

**The "Context Cliff" Phenomenon.** Beyond overall performance, our benchmark provides critical insights into how different model paradigms handle conversational history. To investigate this, we varied the context window duration ($w$) from 0s to 30s and plotted the relative performance change, as shown in Figure 5. Our analysis reveals a stark divergence between paradigms. We find that cascaded models are generally effective at leveraging longer dialogue contexts for improved performance. In contrast, end-to-end models exhibit a significant **"context cliff"** phenomenon: specifically, their performance in IQ and EQ-L. initially improves with more context but then **sharply declines** after an optimal point (around 10-15s). This suggests that current end-to-end architectures suffer from a systemic failure in processing extended interactions, where overly long contexts become a burden rather than an aid. Interestingly, for EQ-P. we observe a more consistent trend across all models: performance generally improves with increased context length, indicating that longer context aids in the generation of more natural and affectively appropriate speech.

## 6 CONCLUSION

In this work, we introduced **Partner-Bench**, the first comprehensive benchmark to address a critical gap: the synthesis of Visually-Grounded IQ and Interactive EQ in AVD. We proposed a fine-grained, multi-dimensional evaluation framework and developed a scalable Data Engine, **Partner-DE**, to systematically construct our benchmark from real-world web videos. Our experiments on Partner-Bench reveal significant insights into the current state of SOTA models, including a universal weakness in reasoning, the current superiority of cascaded architectures, and a novel "context cliff" phenomenon that indicates a systemic failure in handling long-range interactions. We believe that by providing a rigorous standard and a powerful diagnostic tool, Partner-Bench will steer targeted improvements and accelerate the development of truly perceptive AI companions.

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

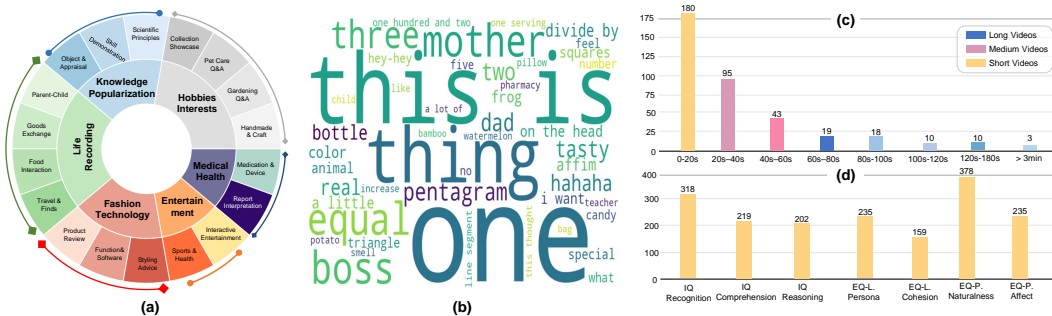

Figure A1: **Dataset Statistics.** (a) Tag Tree for Targeted Video Search. (b) Word Cloud of Dialogue Content. (c) Sample Count by Video Duration. (d) Sample Count by IQ/EQ Dimension.

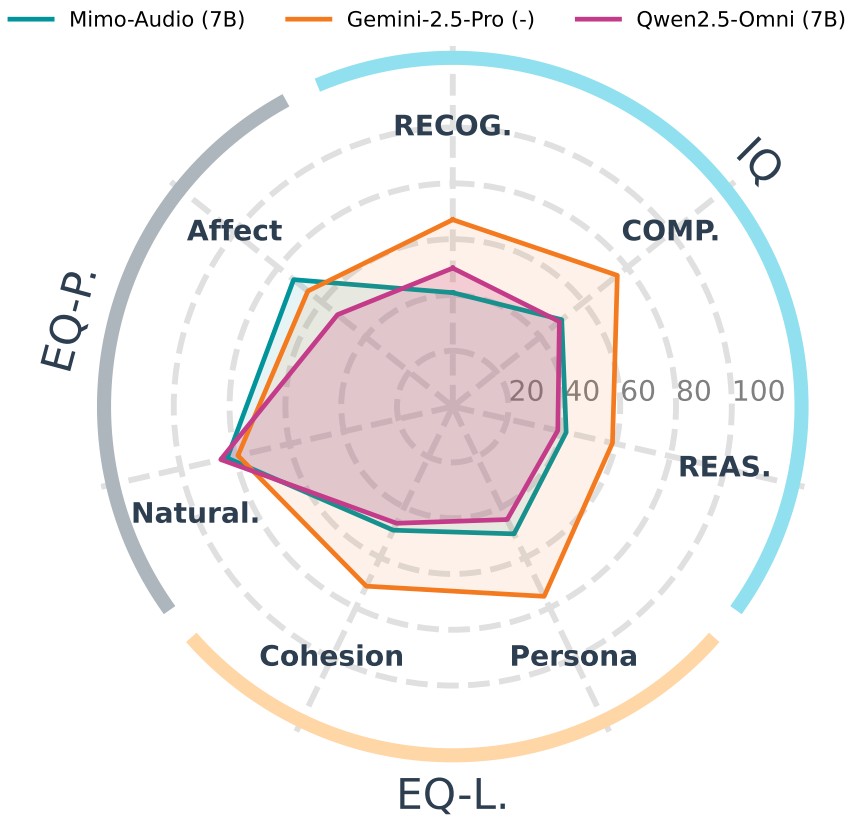

Figure A2: **Comparison between different paradigms across different evaluation dims.**

# A APPENDIX

## A.1 USE OF LLM

I primarily use LLMs to refine my English expressions and organize the overall logic of my paper.

