# OpenReview forum: "Partner-Bench: Evaluating Visually-Grounded IQ and Interactive EQ in Audio-Visual Dialogue"
_ICLR.cc/2026/Conference — ICLR 2026 Conference Withdrawn Submission_

### Official Review · Reviewer_WJjL · 2025-11-01

**Soundness:** 2
**Presentation:** 2
**Contribution:** 2
**Rating:** 2
**Confidence:** 4

**Summary:**

This paper proposed a new audio-visual benchmark focusing on assessing the visual-grounded IQ and interactive EQ. The evaluation methods are not enough.

**Strengths:**

- This benchmark focuses on visual-audio dialogue understanding by assessing the EQ effects.
- It includes “Modality Contrastive Filtering ” in its data creation processes.

**Weaknesses:**

- This paper claims three points I concluded that are different from previous works: (1) visually-oriented, (2) interactive cohesion, (3) EQ-oriented evaluation settings.  However, (2) and (3) are mixed in the benchmark setting which is a little bit confusing.

- The comparison in Figure 1 seems weak, I don’t know why “Audio-Visual Understanding” will output only “description” without “answering”. If providing the visual inputs and the question, I think the proprietary models like gemini-2.5-pro and GPT-5 will give the answer along with the thinking process, this example cannot be suitable to differentiate the “Audio-Visual Understanding” with “Audio-Visual  Dialogue” for readers. In my opinion, I think this benchmark is a visual version of audio dialogue and the interactive is a new setting for assessing the consistent Persona. The authors use passive and proactive methods to differentiate their work.

- Video caption model + ASR -> textresponse + text2speech setting should be included in the evaluation. In this setting, you can assess more video LLMs and you can use asr to represent the audio information. Additionally, it is so confusing the output of video caption model is text format, so the notion in Table 2 “Cascade (Cap2Dial): Video Caption Model w/ Speech-to-Speech Model ” should be “Cascade (Cap2Dial): Video Caption Model w/ Text-to-Speech Model ”. I guess the authors probably got the naming wrong.

- The results in Table 2 seem to indicate the audio information is not important as Video LLM + TTS achieves the best performance, therefore we may require only the video benchmark with interactive EQ setting. It will lower the weight of this benchmark.

- The results show that the settings of this benchmark is not challenging enough.

**Questions:**

Did authors test with current SOTA models on Audio-Visual Understanding? I am not convinced the output is restricted by only providing “description” rather than “answering”.

---

### Official Review · Reviewer_rh6w · 2025-11-01

**Soundness:** 2
**Presentation:** 2
**Contribution:** 3
**Rating:** 2
**Confidence:** 4

**Summary:**

The paper under review introduces a novel benchmark called Partner-Bench for visually grounded Intelligence Quotient (the ability to accurately perceive the visual world) and Interactive Emotion Quotient (the ability to communicate naturally) in audio-visual dialog. The benchmark is related to the shift from tool-oriented assistants (working with tools like search engines) to partner-oriented assistants (working with people). In total, Partner-Bench contains 376 samples with 3.49 hours of content. The content for Partner-Bench was collected from diverse sources such as YouTube and BiliBili.

The authors also introduce a pipeline for data preparation, preprocessing, and filtering for Partner-Bench. The pipeline consists of the following steps:

1. Data sourcing of videos from YouTube and BiliBili
2. Preprocessing audios using Voice Activity Detection models and state-of-the-art ASR models
3. Preprocessing dialogs (formatting into two-party dialogs)
4. Filtering dialogs
5. Modality contrastive filtering and tagging
6. Manual verification to filter out videos that contradict the authors’ standards

As the main evaluation methodology, the authors propose using LLM-as-a-judge with Gemini-2.5-Pro as the annotator. They demonstrate that LLM-as-a-judge correlates with human annotation on 100 dialog units. The overall score aggregates IQ and EQ scores with specific weights. The evaluation includes various models and pipelines. As the field lacks end-to-end models, the authors evaluated several bimodal models in two additional cascade pipelines: cap2dial (a visual/video model describes the environment and a speech2speech model generates the answer) and avqa2tts (an audio-visual model analyzes the input and a TTS model generates the answer).

The main results show that all approaches perform significantly worse than humans, and even the best model underperforms the human baseline. Cascade pipelines perform overall better than end-to-end approaches. Interactive Emotion Quotient remains a challenging task. According to the authors’ evaluation, Gemini-2.5-Pro is the state-of-the-art model on Partner-Bench.

**Strengths:**

A substantive assessment of the strengths of the paper, touching on each of the following dimensions: originality, quality, clarity, and significance. We encourage reviewers to be broad in their definitions of originality and significance. For example, originality may arise from a new definition or problem formulation, creative combinations of existing ideas, application to a new domain, or removing limitations from prior results.

● The authors present a novel audio-visual dialog benchmark for partner-oriented tasks.

● The benchmark generation pipeline is mostly scalable, except for the manual verification step. The methodology could be applied to any audio-video source.

● The authors evaluate different paradigms (cascade / end-to-end) and models.

**Weaknesses:**

A substantive assessment of the weaknesses of the paper. Focus on constructive and actionable insights on how the work could improve towards its stated goals. Be specific, avoid generic remarks. For example, if you believe the contribution lacks novelty, provide references and an explanation as evidence; if you believe experiments are insufficient, explain why and exactly what is missing, etc.


● The authors state that Gemini-2.5-Pro is the top performer on Partner-Bench. However, Gemini-2.5-Pro also serves as the main annotator in the LLM-as-a-judge pipeline for Partner-Bench. It is a known issue that models in LLM-as-a-judge pipelines may prefer their own predictions, which creates a strong bias. This bias could be reduced if the LLM-as-a-judge pipeline consisted of an ensemble of models. Alternatively,
Gemini-2.5-Pro could be removed or replaced with another state-of-the-art proprietary model. The LLM-as-a-judge methodology also requires discussion in the Limitations section, where potential biases could be described.

● Some aspects of the paper are unclear. First, the prompts for the referee model are not included in the paper, though they are important to validate objective evaluation. The description of specific models used in the data mining pipeline is also missing (for instance, which state-of-the-art ASR model was used in line 319?). The human verification step could be elaborated: it is unclear which standards of video quality were followed, how the appropriateness of questions was assessed, and how many human experts participated.

● Using LLM-as-a-judge requires strong justification of its correlation with human evaluation. The authors report some comparison between human annotation and the LLM referee, but it should be expanded. A description of the human annotators (how many, whether they are native speakers, etc.) and correlation metrics should be added. Currently, it is difficult to analyze the difference between human and LLM evaluations using only the visualization in Figure 4.

● While the authors state that cascade pipelines perform better than end-to-end models, they use proprietary Gemini models in the cascade pipelines, while the end-to-end setup includes only open-source 8B models. Therefore, the comparison is not fair. This issue could be resolved by including proprietary models in the end-to-end pipelines as well.

**Questions:**

Please list up and carefully describe any questions and suggestions for the authors. Think of the things where a response from the author can change your opinion, clarify a confusion or address a limitation. This is important for a productive rebuttal and discussion phase with the authors.

● Why are only RECOG and COMP scores bolded in Table 2? Bold all best results and underline the top-2. It would greatly help to understand the results of evaluation.

● What is the purpose of converting the 5-point scale into 100?

● The word cloud in the appendix could be replaced with a more interpretable visualization.

---

### Official Review · Reviewer_17kE · 2025-11-01

**Soundness:** 3
**Presentation:** 3
**Contribution:** 3
**Rating:** 4
**Confidence:** 3

**Summary:**

This paper tackles the evaluation of Audio-Visual Dialogue (AVD) systems that must combine Visually-Grounded IQ (intelligence in reasoning over visual+audio content) with Interactive EQ (empathic, natural conversational ability). The authors argue that prior benchmarks only address one of these aspects in isolation – audio dialogue tests measure interactivity (EQ) but ignore visual context, while audio-visual understanding tests measure visual reasoning (IQ) but lack interactive dialogue. To bridge this gap, the paper introduces Partner-Bench, the first benchmark explicitly designed to evaluate the synthesis of both IQ and EQ in AVD models. Partner-Bench is constructed via a novel data pipeline and provides a fine-grained 7-dimensional evaluation framework covering multiple sub-skills of IQ and EQ. The benchmark consists of 376 multi-turn audio-visual dialogue samples drawn from real web videos, each annotated for the specific IQ/EQ challenges it poses.

**Strengths:**

The main strengths of the paper are as follows:

1. There is a significant gap in the interconnected three- and more-modality benchmarks, while prior audio dialogue is visually blind; audio-vision understanding is not interactive, Partner-Bench evaluates both together and covers the gap.
2. The authors provide a detailed and transparent data-curation pipeline. Especially interesting is the use of hierarchical topic trees for video retrieval and the dual-speaker normalization to enforce structured dialogue make the process reproducible and analytically clean.
3. Partner-Bench simulates a real-life conversational context: the model must assume a speaker role and respond coherently given both dialogue history and audiovisual evidence that makes this synthetic data more real-life.
4. The benchmark evaluates a diverse lineup of state-of-the-art models (cascaded and end-to-end, from small (7B) to very large ones) and includes a human baseline.

**Weaknesses:**

The main weaknesses of the paper are as follows:

1. The evaluation relies heavily on Gemini-2.5-Pro as the sole scoring model. While the authors conduct a correlation test against human ratings, Gemini also serves as a component within the top-performing cascaded systems, essentially grading outputs influenced by itself. Thus, still there are concerns about potential self-evaluation bias or stylistic preference toward responses similar to Gemini's own generation patterns. Although the alignment with human scores is reassuring, it would be safer to validate results with an independent judge or multiple scoring models to ensure full impartiality and reproducibility.
2. Partner-Bench contains only 376 curated dialogue units (with two main data sources), which may be insufficient to capture the full diversity of real-world audio-visual dialogue. A small benchmark risks overfitting of models to particular types of conversational or visual patterns and limits statistical robustness for cross-model comparisons.

**Questions:**

I have the following questions to the authors:

1. Since Gemini-2.5-Pro is used both as a scorer and as part of the best cascaded systems, did you test evaluation stability with an independent judge (e.g., GPT-series)? How consistent are rankings when a different referee is used? 2. What is the approximate language and topical distribution of the dataset?
3. How was the human baseline obtained, did multiple annotators participate, and how consistent were their ratings across the seven dimensions?
4. How sensitive are the final rankings to the 0.5/0.25/0.25 weighting scheme between IQ, EQ-L, and EQ-P? Did you test alternative weighting strategies?
5. Have you analyzed what specifically causes the drop in long-context performance: input length limits, attention decay, or dialogue inconsistency? Would summarization or retrieval-based memory mitigate it?
6. Have you measured how an audio-only variant of each system performs on Partner-Bench to quantify the gain provided by visual grounding?
7. Could you please provide the main statistics on the data source, specifically the main language, and what specific ASR models and other pipeline components were used for data curation?

If the authors address these questions, I am willing to increase my score.

---

### Official Review · Reviewer_JPwY · 2025-11-03

**Soundness:** 3
**Presentation:** 3
**Contribution:** 3
**Rating:** 4
**Confidence:** 5

**Summary:**

This paper introduces Partner-Bench, a novel benchmark designed to evaluate Audio-Visual Dialogue (AVD) models by synthesizing two critical capabilities: Visually-Grounded Intelligence Quotient (IQ) and Interactive Emotional Quotient (EQ). The authors highlight a significant gap in existing evaluation methods, which either assess IQ or EQ in isolation but fail to measure their integration. To address this, they propose a fine-grained, 7D evaluation framework and develop Partner-DE, a scalable data engine that automatically mines and annotates high-quality conversational data from web videos. The benchmark comprises 376 samples and offers a rigorous protocol for diagnosing model weaknesses, such as the "context cliff" phenomenon. The work aims to accelerate progress towards developing perceptive and engaging AI companions.

**Strengths:**

Several strengths of the proposed paper can be highlighted:
1. Novelty: Partner-Bench is the first benchmark to unify the evaluation of Visually-Grounded IQ and Interactive EQ in AVD, addressing a critical methodological gap.
2. Comprehensive Framework: The seven-dimensional evaluation (e.g., Recognition, Reasoning, Persona, Naturalness) enables granular diagnostics beyond monolithic scores.
3. Scalable Data Engine: Partner-DE demonstrates innovation in automating data mining, filtering, and annotation, reducing reliance on costly manual curation.
4. Rigorous Evaluation: The model-as-judge protocol using Gemini-2.5-Pro shows strong correlation with human judgments, ensuring scalability and reliability.
5. Actionable Insights: Experiments reveal key findings, such as the performance gap between cascaded and end-to-end models and the "context cliff" phenomenon.

The value of Partner-Bench lies in its holistic approach to evaluating AVD models, which is currently absent in the literature. The benchmark’s fine-grained dimensions enable targeted diagnostics, while the data engine demonstrates scalability and practicality. The experiments reveal critical limitations of state-of-the-art models, such as their struggle with reasoning and long-context interactions, offering clear directions for improvement. These contributions are likely to influence the development of next-generation AI systems and foster interdisciplinary collaboration between vision, language, and speech communities.

**Weaknesses:**

However there are several weak points I'd like to mention:
1. Dataset Scale: With only 376 samples (3.49 hours), the benchmark may lack diversity and robustness compared to larger datasets.
2. Dependence on Proprietary Models: The use of Gemini-2.5-Pro for annotation and evaluation limits reproducibility and accessibility for the broader community.
3. Limited Model Diversity: Evaluated models are predominantly from industry (e.g., Gemini, Qwen), with fewer open-source or academic alternatives.
4. Context Cliff Analysis: While identified, the underlying causes of the "context cliff" are not deeply investigated (e.g., architectural limitations or training data biases).
5. Why GPT-5, Claude are not in a comparison list? It seems more models should be evaluated for a proper comparison. E.g., take some leaderboard and try to evaluate the best perception models
6. There is not much attention paid to reasoning capabilities of the models - how this property affects quality?

**Questions:**

1. How do you plan to expand Partner-Bench in terms of scale, diversity, and task complexity to ensure long-term relevance?
2. Can you provide more analysis on the "context cliff" phenomenon? For instance, is it related to model architecture, training data, or attention mechanisms?
3. Have you explored using open-source models (e.g., LLaVA, Whisper) in Partner-DE to improve reproducibility?
4. How does the performance of cascaded models change when using weaker captioning or TTS components? Is their superiority robust to component choices?

---

### Note · Authors · 2025-11-12

I have read and agree with the venue's withdrawal policy on behalf of myself and my co-authors.